# RUNX1/CEBPA Mutation in Acute Myeloid Leukemia Promotes Hypermethylation and Indicates for Demethylation Therapy

**DOI:** 10.3390/ijms231911413

**Published:** 2022-09-27

**Authors:** Ekaterina I. Romanova, Anatoliy V. Zubritskiy, Anna V. Lioznova, Adewale J. Ogunleye, Vasily A. Golotin, Anna A. Guts, Andreas Lennartsson, Oleg N. Demidov, Yulia A. Medvedeva

**Affiliations:** 1Institute of Bioengineering, Research Center of Biotechnology, Russian Academy of Science, 117312 Moscow, Russia; 2Department of Biomedical Physics. Moscow Institute of Physics and Technology, 141701 Dolgoprudny, Russia; 3Institute of Cytology, RAS, Laboratory of Molecular Medicine, 194064 St. Petersburg, Russia; 4State Research Institute of Lake and River Fisheries, 199053 St. Petersburg, Russia; 5INSERM UMR1231, University of Burgundy Franche-Comté, 21078 Dijon, France; 6Institute of Pharmacy, Sechenov University, 119435 Moscow, Russia; 7Department of Biosciences and Nutrition, Karolinska Instituten, 141 83 Huddinge, Sweden; 8Sirius University of Science and Technology, 354340 Sochi, Russia

**Keywords:** AML, DNA methylation, RUNX1, CEBPA, TET2, epigenetics, BIK

## Abstract

Acute myeloid leukemia (AML) is a rapidly progressing heterogeneous disease with a high mortality rate, which is characterized by hyperproliferation of atypical immature myeloid cells. The number of AML patients is expected to increase in the near future, due to the old-age-associated nature of AML and increased longevity in the human population. RUNX1 and CEBPA, key transcription factors (TFs) of hematopoiesis, are frequently and independently mutated in AML. RUNX1 and CEBPA can bind TET2 demethylase and attract it to their binding sites (TFBS) in cell lines, leading to DNA demethylation of the regions nearby. Since TET2 does not have a DNA-binding domain, TFs are crucial for its guidance to target genomic locations. In this paper, we show that RUNX1 and CEBPA mutations in AML patients affect the methylation of important regulatory sites that resulted in the silencing of several RUNX1 and CEBPA target genes, most likely in a TET2-dependent manner. We demonstrated that hypermethylation of TFBS in AML cells with RUNX1 mutations was associated with resistance to anticancer chemotherapy. Demethylation therapy restored expression of the RUNX1 target gene, BIK, and increased sensitivity of AML cells to chemotherapy. If our results are confirmed, mutations in RUNX1 could be an indication for prescribing the combination of cytotoxic and demethylation therapies.

## 1. Introduction

Acute myeloid leukemia (AML) is a hematologic malignancy characterized by the proliferation of immature myeloid progenitors and substantial genetic, cytogenetic, and epigenetic abnormalities. Recent cohort studies have shown that about 45% of AML patients are characterized by a normal karyotype [1] and the presence of a few core driver mutations [2] in transcription factors and epigenetic regulators (DNMTs, TET2, ASXL1 and IDH1/2 [3]). Cytotoxic drug Cytarabine (Ara-C), in combination with etoposide or alone, remains the first line of treatment for AML patients. DNA-Ademethylating drugs Decitabine and Azacitidine supplement cytotoxic therapy of AML with mutations in epigenetic regulators [4]. Recently, new drugs have been approved for use, including Enasidenib and Ivosidenib—inhibitors targeting recurrent mutations in IDH1/2 genes, respectively, thus extending the therapeutic landscape for AML [5,6].

RUNX1 and CEBPA, transcriptional factors essential for normal hematopoiesis, are also frequently mutated in AML [7]. Point mutations in RUNX1 are detected in 6–33% of cytogenetically heterogeneous AML patients [8,9,10,11]. RUNX1 mutations are almost mutually exclusive to AML, with recurrent genetic abnormalities, and they co-occur with a complex pattern of gene mutations, frequently involving mutations in epigenetic modifiers (ASXL1, IDH2, KMT2A, EZH2). RUNX1-mutated AML is associated with distinct clinicopathologic features and inferior prognosis, depending on a spectrum of co-occurring mutations [12]. CEBPA mutations occur in about 7–15% of AML cases, and most of them are double (bi-allelic) mutations [13,14]. Biallelic CEBPA mutation (biCEBPA) in AML with a normal karyotype is distinguished as a distinct AML subtype and is associated with a favorable clinical outcome [15,16,17]. Monoallelic CEBPA-bZip mutations are also associated with favorable prognosis in children [18].

TET enzymes play a major role in active DNA demethylation [19]. TET2, which is commonly mutated in AML [20], does not have a DNA binding domain, and requires molecular partners—usually transcriptional factors (TFs)—to guide it to specific genomic locations [21,22,23,24,25]. In normal conditions, both RUNX1 and CEBPA could contribute to attracting TET2 to their binding sites (TFBS), causing demethylation of regulatory regions and keeping the corresponding genes active [22,23,25,26].

We hypothesized that this mechanism might be disturbed in AML in patients with RUNX1 or CEBPA mutation. We presume that the lack of RUNX1/CEBPA may reduce TET2-induced demethylation near RUNX1/CEBPA binding sites, indirectly affecting the DNA methylation of regulatory regions and keeping RUNX1 and CEBPA-regulated genes inactive. In line with this hypothesis, we show that chemical demethylation improves sensitivity of such cells to state-of-the-art AML chemotherapy, at least partially through activation of the pro-apoptotic gene BIK. 

## 2. Results

### 2.1. Mutations in RUNX1 and CEBPA Lead to Hypermethylation near Their Binding Sites and Change of Expression of Regulated Genes

For this study, we used data on DNA methylation and gene expression from The Cancer Genome Atlas (TCGA) AML cohort of 186 patients. Of 186 patients, 17 (9%) and 13 (7%) patients had mutations in RUNX1 and CEBPA, respectively. The types of the mutations are described in Appendix A. In patients with RUNX1/CEBPA mutation, 5/1 patients had nonsense mutations and 4/8 patients had frameshifts, respectively. These mutations most likely affect the protein structure or function. The rest of the mutations were missense or in-frame insertions or deletion, the effects of which, on the protein function, could be subtle. We split patients in two groups and compared DNA methylation levels in patients with and without a corresponding mutation. 

CpGs positions within close proximity to RUNX1/CEBPA binding sites (TFBS) showed an increase in DNA methylation in AML patients with corresponding mutations in comparison to patients without mutations (Figure 1A,B,D,E,G,H). For the majority of CpGs, only a mild gain in methylation in RUNX1-mutated AML patients was observed (Figure 1A, Appendix A, average Δmeth = 0.06); conversely, for the differentially methylated CpG in close proximity to RUNX1 TFBS, the gain in DNA methylation was significantly higher (Figure 1A, Appendix A, average Δmeth = 0.18). A similar but less significant tendency was observed for the patients with mutations in CEBPA: the gain in DNA methylation genome-wide was significantly lower than that near CEBPA TFBS (Figure 1G, Appendix A, average Δmeth = 0.09 vs. average Δmeth = 0.13). This tendency did not occur for CEBPA TFBS near hypermethylated CpG TL, but the difference was insignificant, due to a low number of such CpGs (Figure 1J,K).

CEBPA mutation leads to a higher level of genome-wide hypermethylation (Figure 1G, gray dots, average Δmeth = 0.09) when compared to RUNX1 mutation (Figure 1A, gray dots, average Δmeth = 0.06). In the case of RUNX1 mutation, genome-wide hypermethylation was almost lost for CpG TL (Appendix A, gray violines)—functional CpG sites associated with changes in gene expression (see Methods). However, in the case of CEBPA mutation, CpG TL tended to demonstrate hypermethylation independent of the presence of CEBPA TFBS (Appendix A, gray violines). The mechanism behind genome-wide hypermethylation of functional CpGs in patients with CEBPA mutation is unclear, since the majority of genomic CpGs do not have CEBPA TFBS nearby. This suggests that CEBPA regulation of the associated genes might be indirect or specific to a subpopulation of patients, as shown in the work of Figueroa [27].

On average, genes associated with CpGs that dramatically change DNA methylation (strong CpG TL) located near RUNX1 were significantly downregulated (Appendix A). On the contrary, CEBPA mutation leads to downregulation of only a few genes. We focused on genes with the most significantly changed methylation and expression levels in the case of TF mutation (FC > 2, FDR < 0.05, absolute expression value > 0.5). We found 12 and 11 genes that meet these criteria for AML patients with RUNX1 and CEBPA mutation, respectively (Appendix A). For CEBPA mutation, these genes are HOXA10, HOXA9, PTRF, TNS3, FSTL1, GPR109B, PI4K2A, ECE1, LOC283663, SCHIP1, and MFSD2A. Only two of these genes (HOXA9, HOXA10) show general prognostic significance. Hypermethylation of HOXA9 and HOXA10, independent of the patients’ mutation profile is linked to a significantly improved survival rate (Appendix A). For RUNX1 mutation, these genes are BIK, OSBPL5, LGALS3BP, KRT18, CACNA2D4, C20orf197, HOXB3, TNFRSF10C, C10orf91, VSTM1, ZBTB16, and C16orf93. Downregulation of OSBPL5 has been previously reported in a subtype of AML [28], while downregulation of BIK was observed in multiple cancers (reviewed in [29]). Hypermethylation of BIK is associated with a worse survival rate in the long run (t > 500 days, *p*-value = 0.002); however, for the overall survival rate, the difference is insignificant, due to a very similar probability of short-term survival (Appendix A).

Methylation levels of strong CpG TL in AML patients with RUNX1/CEBPA mutation resemble those in immature cells observed in normal hematopoiesis (Figure 1F,L). Cell type deconvolution analysis [30] showed a significant decrease in peritoneal macrophages in AML patients with both mutations and in monocytes in the case of CEBPA mutation (Appendix A). This observation suggests that demethylation of RUNX1 and CEBPA binding sites is required for normal myelopoiesis, and that this program is disrupted in AML with RUNX1 or CEBPA mutations.

### 2.2. TET2 Is Likely Involved in RUNX1/CEBPA TFBS Demethylation in AML

In a recent study, Suzuku et al. showed, through co-immunoprecipitation, that RUNX1 and CEBPA directly interacts with TET2 and, in this way, could recruit it to their TFBS in human cell lines [23]. We hypothesized that this mechanism might be implicated in AML patients with RUNX1 and CEBPA mutation. We presume that the lack of fully functional RUNX1/CEBPA may reduce TET2-induced demethylation of nearby RUNX1/CEBPA binding sites, indirectly affecting the DNA methylation profile of the patients and keeping RUNX1 and CEBPA-regulated genes inactive. 

To verify this hypothesis, we used the TET2 profile determined by ChIP-seq (see Methods for the details). Indeed, the amount of TET2 is significantly increased in close proximity of RUNX1 and CEBPA TFBS in cells with intact RUNX1 and CEBPA (Figure 1C,I), supporting the suggestion that TET2 could be involved in demethylation of these CpGs in normal conditions. Several genes regulated by RUNX1 or CEBPA demonstrated a significant increase in DNA methylation and a decrease in expression in AML patients with RUNX1 or CEBPA mutation, respectively (Appendix A). 

Next, we focused on genes with the most significantly changed methylation and expression levels in the case of TF mutation (FC > 2, FDR < 0.05, absolute expression value > 0.5). BIK/OSBPL5 and HOXA9 were among those genes in the case of RUNX1 and CEBPA mutation, respectively. ChIP-qPCR confirmed reduced TET2 presence in OSBPL5 and BIK genes in the OCI-AML5 cell line (a line with a reported RUNX1 mutation (Figure 2A,B)), and in HOXA9 in the Kasumi-6 cell line (a line with a reported CEBPA mutation (Appendix A)), supporting the role of TET2 in TFBS demethylation. 

### 2.3. Demethylation Treatment Restores the Sensitivity of RUNX1-Mutated AML Cells to Ara-C

Cytarabine (Ara-C), in combination with etoposide or alone, remains the first line of treatment for AML patients. AML cell lines with RUNX1 mutations OCI-AML5 and Mono-Mac1 were more resistant to Ara-C than wild-type (wt) RUNX1 AML cell line OCI-AML2 (Figure 2C). To restore the sensitivity of RUNX1-mutated AML cells to chemotherapy, we pretreated AML cells with the demethylating agent azacytidine (AZA). AZA did not potentiate Ara-C-dependent cytotoxicity in RUNX1 wild-type OCI-AML2 cells, but significantly reduced the viability of AML cells bearing mutant RUNX1, OCI-AML5, and MonoMac1 (Figure 2D). Demethylation significantly potentiated the cytotoxic effect of chemotherapy in AML with RUNX1 mutations. Moreover, treatment with a demethylating agent increased the expression of pro-apoptotic gene BIK, both at mRNA and protein levels, which can explain the sensitization of RUNX1-mutated leukemic cells to chemotherapy by Ara-C (Figure 2E,F).

## 3. Discussion

One of the distinctive features of AML is a significant disturbance in epigenetic regulation. Mutations in IDH1/2, TET2, and DNMT3A are known to be early events in the development of AML and can be detected years before diagnosis [31,32]. We presented evidence that in AML patients, TET2 is specifically involved in the DNA demethylation near TFBS of RUNX1 and CEBPA, particularly in demethylation of CpG TL—functional CpG positions in which methylation is a marker of the gene expression nearby. Thus, a lack of fully functional RUNX1 or CEBPA proteins in AML patients with a corresponding mutation could prevent the interaction of TET2 with the regions around their TFBS, leading to methylation of regulatory regions and suppression of several genes’ expressions.

Regulatory regions of RUNX1/CEBPA target genes were affected by lack of TET2 and, as a result, demonstrated extensive hypermethylation. Changes in DNA methylation followed different scenarios in patients with mutations of both types: in the case of RUNX1 mutation, the most pronounced changes happened in CpG TL near RUNX1 TFBS, while, in case of CEBPA mutation, we observed a DNA methylation change in genome-wide CpG TL, independent of the presence of CEBPA TFBS, supporting the idea that CEBPA may not only affect DNA methylation through attraction of TET2. Recently, it has been shown that CEBPA interacts with DNMT3A N-terminus and, in this way, blocks DNMT3A from accessing DNA substrate, thereby inhibiting its activity [33]. Thus, this suggests that regulation of DNA methylation by CEBPA has a complex nature.

However, we were able to detect only a few genes that respond to hypermethylation caused by CEBPA mutation. Among those genes whose expression is strongly affected by CEBPA mutation, we identified HOXA9 and HOXA10. We showed that hypermethylation of HOXA9 and HOXA10 genes was linked to significantly improved rates of long-term survival, most likely indirectly contributing to the long-term survival rates detected in patients with double CEBPA mutations [17]. Thus, methylation levels of HOXA9/HOXA10 could be considered prognostic markers in AML.

It has previously been shown [8] that patients with RUNX1 mutation demonstrate lower survival rates. Changes in TET2-dependent levels of DNA methylation close to RUNX1 TFBS in AML patients with a corresponding mutation lead to the repression of multiple regulated genes, including pro-apoptotic gene BIK. Although patients with both RUNX1 and CEBPA mutations demonstrated the most significant increase in DNA methylation near their TFBS in CpG TL—functional CpG positions whose methylation correlates to the expression of a nearby gene—the downstream effect on gene expression of RUNX1 is more pronounced. This observation is in line with the more severe consequences of RUNX1 mutation on patients’ survival.

The chromosomal translocation t(8;21)(q22;q22), generating the RUNX1/RUNX1T1 fusion gene, is the most prevalent chromosomal rearrangement in AML, with an incidence rate of about 15% in children and young adults [34]. The translocation produces a fusion protein composed of the RUNX1 DNA-binding Runt domain and the almost complete open reading frame of RUNX1T1. Since the resulting fusion protein lacks a TET2 binding domain, we believe that such a rearrangement should prevent the fusion protein from attracting TET2 to its TFBS and demethylating the regions [35,36]. This observation is supported by the epigenetic heterogeneity in patients with RUNX1/RUNX1T1 fusion [37]. 

We used the OCI-AML5 cell line as a proxy for AML cells with a RUNX1 mutation. However, this cell line also has 41 more mutated genes, including TET2 mutation (S825*) [38]. A more accurate study of the mutations in OCI-AML5 cell line shows bi-allelic mutations S825*/Y1148C in the TET2 gene [39]. It is not entirely clear how all these mutations may affect TET2-dependant regulation. Since one allele of TET2 in OCI-AML5 contains only a substitution to relatively similar amino acids, we hypothesize that levels of functional TET2 protein could be partially reduced, but not totally eradicated. However, global DNA methylation levels are not changed in this cell line [40], suggesting that even partially functional TET2 can perform demethylation. Although the evidence is indirect, we believe that the changes we observed in RUNX1 target genes in OIC-AML at least partially occur due to RUNX1 mutation. 

Similarly, we used the Kasumi-6 cell line as a cell line with a CEBPA mutation. However, this cell line also has RUNX1 overexpressed. It has been previously shown [23] that overexpression of RUNX1 leads to the increased levels of TET2 near RUNX1 binding sites. This is despite RUNX1 and CEBPA biding sites being quite different (Appendix A), so it is highly unlikely that increased binding of RUNX1 will affect binding of CEBPA in any way, or that it will increase the TET2 signal near CEBPA TFBS. Even if, in some cases, there is a cooperative RUNX1-CEBPA binding, the increased binding of RUNX1 should have an increased TET2 signal; however, we observed a decreased TET2 signal in the KASUMI-6 cell line, supporting the hypothesis that overexpression of RUNX1 should not interfere with CEBPA mutation-related effects. 

Due to a lack of TET2 ChIP-seq data in AML-related cells, our results on genome-wide TET2 distribution should be considered with caution. We managed to support these results for several genes with ChIP-qPCR (Figure 2A,B, Appendix A), suggesting that genome-wide distribution of TET2 in MCF cells could be considered a decent proxy for the TET2 distribution in other cells. 

Our results indicate that methylation patterns are significantly changed not only in AML patients with mutations in epigenetic regulators, but also in AML patients carrying mutations in RUNX1 or CEBPA. Currently, mutations in epigenetic regulators, such as IDH1/2 and DNMT3A, in AML patients serve as molecular predictors of a good response to therapy with hypomethylating agents (HMA) such as Azacitidine or Decitabine. Recently, a therapeutic strategy for the treatment of CEBPA-mutated leukemia with DNA-hypomethylating agents has been suggested. 

Moreover, we demonstrated the reactivation of the expression of pro-apoptotic protein BIK (Bcl-2-interacting killer) by HMA in AML cells. We also showed that hypermethylation of BIK might have an effect on long-term patient survival. This is in line with the previously proposed idea that therapeutic approaches to activate the pro-apoptotic BH3-only genes, including BIK, might improve the clinical outcome of chemotherapy treatments in drug-resistant AML [29]. BIK-associated mechanisms may be partially responsible for the observed complementary effect of the recently FDA-approved protocol for the treatment of AML in elderly patients, which implements a combination of Bcl-2 inhibitor Venetoclax and HMA [41]. Our data advocates for the rationale of prescribing epigenetic therapy with hypomethylating agents for the subtypes of AML with mutations in transcriptional factor RUNX1, although this hypothesis would need to be confirmed in a prospective AML cohort.

## 4. Materials and Methods

### 4.1. Patients Data

Mutations, DNA methylation (Illumina 450k Array beta-values) and gene expression (RNA-seq, RPKM) profiles for the bone marrow of 186 AML patients were obtained from The Cancer Genome Atlas (TCGA) [42]. Methylation beta-values (Illumina 450k array) were obtained by the TCGA consortium. DNA methylation data were generated using the ‘EGC.tools’ R package (version 1.3.0) after processing raw IDAT files for each sample with the ‘methylumi’ R package (version 2.3.22) by the TCGA consortium. 

### 4.2. Additional Data

DNA methylation profiles in different stages of normal granulopoiesis (Illumina 450k beta-values) were obtained from FACS-sorted bone marrow cells of voluntary healthy donors [43]. The data were preprocessed with the Genome Studio module 1.8. by [43]. Genome-wide TET2 location profiles (ChIP-Seq) in the MCF7 cell line were obtained from the work of Wang and colleagues [44]. 

### 4.3. TFBS Prediction

RUNX1 and CEBPA TFBS were annotated in all CpG-centered 200bp regions (+/−100 bp) using positional weight matrices (PWM) from HOCOMOCO v11 (*p*-value < 0.001) [45]. We predicted 53,443 sites for RUNX1 and 47,157 for CEBPA. To reduce the number of false positives, we filtered out those predicted TFBS that did not co-locate with the ChIP-seq peak for a corresponding TF from Cistrome (A, B and C categories only) [46]. As a result, we obtained 10570 sites for RUNX1 and 5263 for CEBPA, respectively. 

### 4.4. Statistical Analysis

To determine CpG sites critical for gene regulation, we used a methodology similar to the one we recently published [47]. Briefly, for each 336478 CpG associated with a gene (Illumina 450k Array annotation), we calculated Spearman correlation coefficient (SCC) between the CpG methylation and the expression profile across all patients. A CpG dinucleotide with a significant SCC (FDR < 0.005, Benjamini–Hochberg method (BH)) was considered critical for gene regulation and referred to as a CpG Traffic Light (CpG TL). In this way, we detected 25928 CpG TLs. We defined strong CpG TLs as those CpG TLs that not only match the criteria of CpG TL, but also demonstrated a significant change in gene expression and DNA methylation (see below) in AML patients with vs. without a mutation in RUNX1/CEBPA gene, respectively. 

We detected differentially methylated CpGs (DM CpG) between patients with a mutation in RUNX1/CEBPA and those without a mutation (Student’s *t*-test, FDR < 0.05, BH). We considered a DM CpG as differentially hypermethylated CpG (DHM CpG) if the mean difference in AML patients with RUNX1/CEBPA mutation and those without it was greater than 0.15. Differentially expressed genes (DEG) were selected in a similar way (Student’s *t*-test, FDR < 0.05, BH). TET2 ChIP-Seq analysis was performed with Deeptools [48].

Detection of a cell type composition was performed with an Epidish package [30]. To cover AML cell types more accurately, the default set of DNA methylation profiles was supplemented with the DNA methylation profiles of mononuclear cells from AML patients [49] and immature blood cells [43].

### 4.5. ChIP-qPCR

Chromatin immunoprecipitation (ChIP) was performed as described previously [50] with the antibody to TET2 in OCI-AML2 cells (wt RUNX1) and OCI-AML5 (mutated RUNX1). Eighteen million AML-2 or AML-5 cells were washed twice with ice-cold PBS and resuspended in 6 mL of ice-cold PBS. The cells were first fixed with 2 mM DSG for 30 min with rotation at room temperature (RT), after which 37% formaldehyde was added to achieve a final concentration of 1%, and samples were rotated for another 10 min. Glycine was added to a final concentration of 125 mM, and the fixation was quenched for 5 min at RT. Fixed cells were washed twice with ice-cold PBS and resuspended in the Sodium dodecyl sulfate (SDS) buffer. After 10 min rotation at RT, cells were spun down for 2 min at 12,000 RCF, resuspended in the IP buffer and sonicated with Bioruptor (Diagenode). Upon sonication, chromatin was diluted with an SDS-free IP buffer to reach the final SDS concentration of 0.1%. Cellular debris was cleared by centrifugation at 16,900 RCF (4 °C) for 20 min, leaving a chromatin-rich suspension. After centrifugation, the supernatant was divided into input (1% vol), IP (67% vol) and mock (32% vol). The IP sample was incubated with 1ug anti-TET2 (N-term) Rb polyclonal antibody (ab230358) for 18 h with rotation at 4 °C. Mock samples were incubated with no antibody under the same conditions. Next, the IP and mock samples were incubated for 3 h with Protein G SureBeads (BioRad, Hercules, CA, USA), pre-blocked with BSA and salmon sperm DNA, with rotation at 4 °C. Beads were washed 3 times with a low-salt buffer, 2 times with a high-salt buffer, 1 time with an IP buffer containing 0.1% SDS. Protein G beads were resuspended in the decrosslinking buffer and incubated overnight at 65 °C with Proteinase K treatment. After decrosslinking, the DNA was extracted with the Qiagen MinElute kit. For the primers sequences used for qPCR, see Appendix A.

### 4.6. Cell Treatment 

Ara-C (20 μm; Cytosar, Astellas Pharma Inc., Tokyo, Japan) were added to experimental wells after 48 h of cultivation. To test the combination therapy with AZA (3 μM, decitabine, Sigma-Aldrich, MO, Saint-Louis, USA), cells at a concentration of 1 million per ml were precultured for 48 h with 3 μM AZA. Thereafter, Ara-C was added at a concentration of 12.8 μM, and culture was continued for an additional 72 h.

### 4.7. Cell Viability Testing 

The CyQUANT™ XTT Cell Viability Assay kit was used to analyze cell viability. To test the residual viability after exposure to cytarabine (Ara-C), 1 million cells per ml were seeded on a 96-well plate with various concentrations of Ara-C (up to 10 μM) and cultured in humid conditions at 37 °C and 5% CO_2_ for 72 h. The viability test was carried out using the XTT reagent spectrophotometrically at a wavelength of 490 nm.

### 4.8. Western Blots

Western blots were performed using anti-BIK antibodies (Cell Signaling, Danvers, MA, USA). 

## Figures and Tables

**Figure 1 ijms-23-11413-f001:**
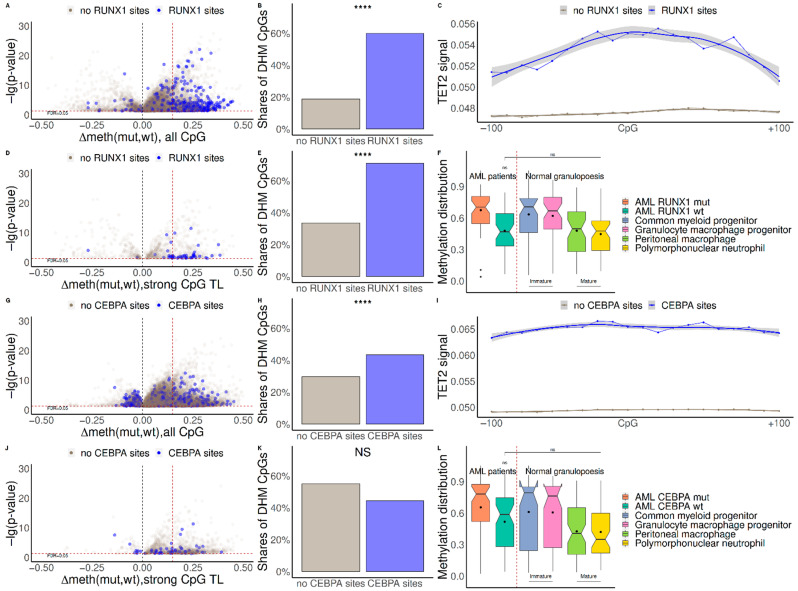
(**A**,**D**,**G**,**J**): Difference in the average levels of DNA methylation between AML patients with and without RUNX1/CEBPA mutation (Δmeth). Differentially methylated CpG in close proximity to TFBS and the rest of the differentially methylated CpGs are represented as purple and gray dots, respectively. Changes in DNA methylation genome-wide and near CpG TL are represented in panels (**A**,**G**) for RUNX1 and (**D**,**J**) for CEBPA, respectively. (**B**,**E**,**H**,**K**): Proportions of hypermethylated CpGs near TFBS (purple) and in the rest of the genome (gray) in patients with mutated RUNX1/CEBPA. Proportions of hypermethylated genome-wide and near CpG TLs are represented in panels (**B**,**H**) and (**E**,**K**), respectively. Significant differences (*p*-value < 10^−7^) are marked by ****, insignificant difference is marked by NS. (**C**,**I**) TET2 binding around hypermethylated CpG near RUNX1/CEBPA TFBS (purple) and in the rest of the genome (gray). (**F**,**L**) Average methylation levels of strong CpG TL near RUNX1/CEBPA TFBS in different stages of normal hematopoiesis and in AML patients with or without the corresponding mutation.

**Figure 2 ijms-23-11413-f002:**
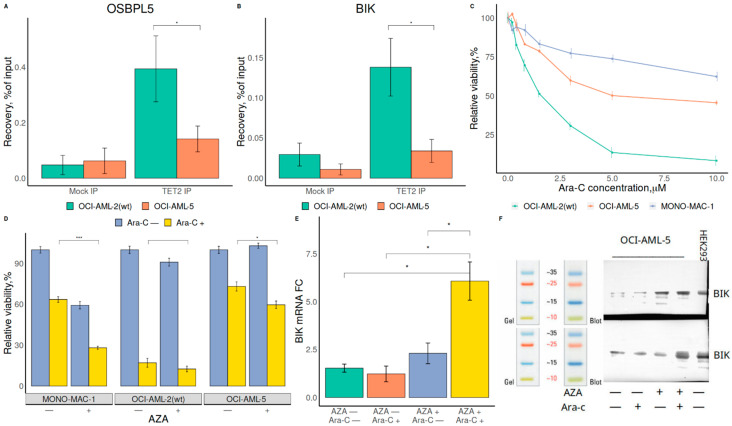
(**A,B**): TET2 binding (ChIP-qPCR) in OSBPL5 (**A**) and BIK (**B**) gene in OCI-AML2 cells (wt RUNX1) and in OCI-AML5 (mutated RUNX1). Significant differences (*p*-value < 0.05) are marked with *. (**C**) Effect of Ara-C on viability of the AML cell lines with and without RUNX1 mutations. (**D**) Effect of the demethylating agent AZA in combination with Ara-C on the viability of AML cell lines with and without RUNX1 mutations. Significant differences (*p*-value < 0.05 and *p*-value < 0.0005) are marked * and *** respectively. (**E**,**F**): Expression of BIK in RUNX1-mutated OCI-AML5 cells before and after treatment with AZA and Ara-C: mRNA level (**E**) and protein level (**F**).

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
