# Peer review of "RUNX1/CEBPA Mutation in Acute Myeloid Leukemia Promotes Hypermethylation and Indicates for Demethylation Therapy"

_ijms, 2022, doi:10.3390/ijms231911413_

Round 1

Reviewer 1 Report

This brief report by Romanova and coworkers is an interesting study with implications for clinical the interventions in AML treatment. The study design and the methodology is scientifically sound. The conclusions drawn from the results support the authors' hypothesis wherein its has been proposed that the certain mutations that inactivate demethylases and tent to increase the hypermethylation on the target sites particularly in reference to  TET2 demethylase and its translocators to the TF binding sites i.e. RUNX1 and CEBPA. The molecular mechanism might represent a principal evidence for advancing the treatment strategies of cytotoxic drugs with an adjuvant of epigenetic modulator. 

The authors are requested to screen the manuscript for minor rephrasing such as:

  1. Instead of using the term " we speculate" the authors can consider "we presume"
  2. Results: page 2: Line 84: Remove "that"
  3. Results: page 2: Line 88: The sentence appears to be incomplete as it ends with " as shown in"

Author Response

This brief report by Romanova and coworkers is an interesting study with implications for clinical the interventions in AML treatment. The study design and the methodology is scientifically sound. The conclusions drawn from the results support the authors' hypothesis wherein its has been proposed that the certain mutations that inactivate demethylases and tent to increase the hypermethylation on the target sites particularly in reference to  TET2 demethylase and its translocators to the TF binding sites i.e. RUNX1 and CEBPA. The molecular mechanism might represent a principal evidence for advancing the treatment strategies of cytotoxic drugs with an adjuvant of epigenetic modulator.

Response: We are grateful to the reviewer for careful reading and useful comments to our paper.

The authors are requested to screen the manuscript for minor rephrasing such as:

  • Instead of using the term " we speculate" the authors can consider "we presume".  Response: Done.
  • Results: page 2: Line 84: Remove "that". Response: Done.
  • Results: page 2: Line 88: The sentence appears to be incomplete as it ends with " as shown in". Response: We modified the corresponding sentence. 

We also performed English spell checking of our paper as suggested by the Reviewer. 

Reviewer 2 Report

AML is characterized by alterations in epigenetic pathways. In this study, authors explore the DNA methylation and TET2 methylcytosine dioxygenase recruitment patterns of the AML samples with mutated RUNX1 or CEPBA transcription factors.

Overall, this manuscript suffers from poor presentation and description of the results. There is also insufficient description of the methods in both the main and supplemental methods. Additionally, there are many strange results and I really question the rigor and quality of the results presented. The figure legends do not describe the figures well.

S. Table 2 and 3 presents gene expression results with RPKM values > 3,000. This is inaccurate. There are columns with poor abbreviations that lack definition (i.e. 'cg' or chrome, what is RPKM RUNX1 wt/mut???)

Many plot axis lack labels

The BIK expression results following treatment is inconsequential and the conclusions are overstated given the poor results provided and lack significance.

The authors should provide a clear and concise description of the patient groups, including the number of patients as well as a better description of the nature of the TF mutations.

It is unclear how many patients belong to wt or mutant groups and whether the TF mutations impaired TF binding.  which makes it difficult to interpret the results and determine the novelty and significance

Author Response

Overall, this manuscript suffers from poor presentation and description of the results. There is also insufficient description of the methods in both the main and supplemental methods. Additionally, there are many strange results and I really question the rigor and quality of the results presented. The figure legends do not describe the figures well. 

Response: We are thankful to the Reviewer for the criticism. We hope we were able to improve our paper to match the Reviewer’s standards. We improved the structure of the Results section, included more details in the Method section and modified Figure legends. The corrections are marked in red. 

S. Table 2 and 3 presents gene expression results with RPKM values > 3,000. This is inaccurate. There are columns with poor abbreviations that lack definition (i.e. 'cg' or chrome, what is RPKM RUNX1 wt/mut???) 

Response: Tables S2 and S3 do not represent RPKM per se but rather the fold change (RPKM ratio) of average expressions in the AML patients with wild type and mutated RUNX1. To avoid further misunderstanding we added the expanded column names to the Tables S2 and S3.

Many plot axis lack labels. 

Response: We added the axis legends to Fig. 1c,f,i,l. and Fig. 2a,b. We also believe that the axes labels for Fig. 1b,e,h,k are self-explanatory and will overload the visualization.   

The BIK expression results following treatment is inconsequential and the conclusions are overstated given the poor results provided and lack significance. 

Response:

We provided an additional experimental validation for the statement about BIK involvement into regulation. We validated the change in expression of BIK with qPCR on the mRNA level. The results provided at Fig. 2(e) clearly show the significance between a combined treatment and the controls. 

We also modified the text as follows:

Novelty: 

For the first time, we analyzed a large number of AML patients for the association of RUNX1 and CEBPA mutations with DNA methylation status of RUNX1/CEBPA target genes.  We provided evidence of TET2-dependent mechanisms for RUNX1/CEBPA target genes demethylation. We discovered that the methylation and silencing of these genes might affect the response to chemotherapy. Our results suggest a new BIK-dependent mechanism of hypomethylating therapy benefits for the treatment of  AML. 

Introduction:

In line with this hypothesis we show that chemical demethylation improves sensitivity of such cells to state-of-the-art AML chemotherapy at least partially through activation of pro-apoptotic gene BIK. 

Results:

Moreover, treatment with a demethylating agent increased the expression of pro-apoptotic gene BIK, both at mRNA and protein levels, that can explain sensitization of  RUNX1-mutated leukemic cells to chemotherapy by Ara-C (Fig. 2e-f).

The authors should provide a clear and concise description of the patient groups, including the number of patients as well as a better description of the nature of the TF mutations. It is unclear how many patients belong to wt or mutant groups and whether the TF mutations impaired TF binding.  which makes it difficult to interpret the results and determine the novelty and significance  

Response: We added a specific section describing patiens data and the number of patients in each group to the Methods. We also provided the description of the mutations in RUNX1 and CEBPA as a Supplementary Table S1 and a summary statistics of the mutations to the Methods.

Reviewer 3 Report

Minor

11)      Brief explanation on the DNA methylation array analysis is needed (software/normalization method). Is this the healthy donor a public dataset too?

22)      Was there any effect with AZA in AML samples with a CEBPA mutation in regards to cytotoxicity?

33)      Figure 2 (a, b, d) are the differences significant.

Author Response

Brief explanation on the DNA methylation array analysis is needed (software/normalization method). Is this the healthy donor a public dataset too?
Response: 

DNA methylation data was obtained from TCGA, we did not perform any postprocessing of this data. Methylation beta-values (Illumina 450k array) were obtained by TCGA consortium.  DNA methylation data were generated using the ‘EGC.tools’ R package (version 1.3.0) after processing raw IDAT files for each sample with the ‘methylumi’ R package (version 2.3.22). Background correction and adjustment to equalize the red/green dye bias across samples was then performed. 

TCGA dataset does not contain healthy donors. Therefore, DNA methylation profiles (Illumina 450k array, beta-values) in different stages of normal granulopoiesis were obtained from FACS sorted bone marrow cells of voluntary healthy donors from a publicly available dataset (Rönnerblad et al. 2014). In the original paper the data was pre-processed with the Genome Studio module 1.8. In the current work the data has not also been postprocessed. 

We added a brief summary of the data processing done by others to the Methods section.   

Was there any effect with AZA in AML samples with a CEBPA mutation in regards to cytotoxicity? 

Response: Since the effect of CEBPA mutation on the DNA methylation and patient survival is controversial, we focused on the RUNX1 mutated cells in testing AZA effects. 

Figure 2 (a, b, d) are the differences significant.

Responce: In Figure A differences are significant. For figure B the extra wet experiment was done with more observations, and now it is significant. In figure D for Ara-C+ in Monomac-1 and OCI-AML-5 the difference is significant, in OCI-AML-5(wt) is insignificant.

Reviewer 4 Report

The investigators report that AML-associated mutations in either RUNX1 or CEBPA affect gene silencing by hypermethylation by failing to recruit the demethylase TET2.  The association is based mostly on the paper by Suzuki et al Blood Advances. However, a relationship, let alone a physical one,  between CEBPA and TET2 is not mentioned. Please clarify or find some other substantiation. 

CEBPA and RUNX1 are involved in myelopoiesis, the former for differentiation commitment and latter earlier in development.  The two affected genes OSBPL5 and BIK are not associated with myeloid differentiation.  Their link with myeloid malignancies remains to be demonstrated, but there needs to be some mechanistic link between its silencing and leukemogenesis. 

The mechanism of action of TET2 being localized to DNA via CEBPA or RUNX1 rests on a single paper from Blood Advances in 2017. that serves as my major concern. A second concern is the effects are on two metabolic and pro-apoptotic genes not associated with myeloid malignancies. 

There should be some discussion and analysis of which RUNX1 and CEBPA mutations affect TET2 binding and recruitment and hypermethylation of BIK or OSBPL5. It would be more relevant to focus on the HOXA9/10 genes. IT would be more novel to identify understudied genes in myeloid neoplasia -- perhaps OSBPL5 -- but there needs to be some mechanistic link between its silencing and leukemogenesis.

Author Response

The investigators report that AML-associated mutations in either RUNX1 or CEBPA affect gene silencing by hypermethylation by failing to recruit the demethylase TET2.  The association is based mostly on the paper by Suzuki et al Blood Advances. However, a relationship, let alone a physical one,  between CEBPA and TET2 is not mentioned. Please clarify or find some other substantiation. 

Responce: 

The work of Suzuki et al Blood Advances report RUNX1 mediated TET2 demethylation ans provide an extensive study of this phenomenon. Later, the same approach was applied to other TFs in the work of Suzuki et al Epigenetics & Chromatin. In this work a screening system for TFs with similar properties to RUNX1 has been introduced. They found several TFs potentially capable of attracting TET2 to their binding sites. CEBPA was among them. Physical interactions of TET2 and CEBPA has been demontrsated in Sardana et al, 2018. We tried to modify the text to make it clearer.  

Abstract:

In this work, we show that RUNX1 and CEBPA mutations in AML patients affect the methylation of important regulatory sites that resulted in the silencing of several RUNX1 and CEBPA target genes, probably in a TET2-dependent manner.

Introduction:

In normal conditions both RUNX1 and CEBPA could contribute to attracting TET2 to their binding sites (TFBS) causing demethylation of regulatory regions and keeping the corresponding genes active (Suzuki, Shimizu, et al. 2017; Suzuki, Maeda, et al. 2017; Sardina et al. 2018)

Results:

To verify this hypothesis we use TET2 profile determined by ChIP-seq (see Methods for the details). Indeed, amount of TET2 is significantly increased in close proximity of RUNX1 and CEBPA TFBS in cells with intact RUNX1 and CEBPA (Fig. 1c,i), supporting that TET2 could be involved in demethylation of these CpGs in normal conditions. Several genes regulated by RUNX1 or CEBPA demonstrate a significant increase in DNA methylation and decrease in expression in AML patients with RUNX1 or CEBPA mutation respectively (Supplementary tables S3,S4). ChIP-qPCR confirms reduced TET2 presense in OSBPL5 and BIK genes in the OCI-AML5 cell line (a line with a reported RUNX1 mutation (Fig. 2a,b)), and in HOXA9 in the Kasumi-6  cell line (a line with a reported CEBPA mutation (Supplementary Fig. S6)) supporting the role of TET2 in TFBS demethylation. 

CEBPA and RUNX1 are involved in myelopoiesis, the former for differentiation commitment and latter earlier in development.  The two affected genes OSBPL5 and BIK are not associated with myeloid differentiation.  Their link with myeloid malignancies remains to be demonstrated, but there needs to be some mechanistic link between its silencing and leukemogenesis. 

Response: We did not find a link between these genes and myeloid differentiation. Yet, we found a link of OSBPL5 with AML and of BIK with a wide range of other cancers. We provide these links below.

We added the following statement to the results:
Down-regulation of OSBPL5 has been previously reported in a subtype of AML (Mondet et al. 2021). While downregulation of BIK was observed in multiple cancers (reviewed in (Chinnadurai, Vijayalingam, and Rashmi 2008)).

The mechanism of action of TET2 being localized to DNA via CEBPA or RUNX1 rests on a single paper from Blood Advances in 2017. that serves as my major concern. 

Response: 

As we mentioned above, our hypothesis about RUNX1 and CEBPA serving as guides for TET2 are based on two papers (Suzuki et al Blood Advances and  Suzuki et al Epigenetics & Chromatin). Physical interaction between TET2 and RUNX1 and between TET2 and CEBPA has been confirmed by Western blot in (Chu et al. 2018) and 

(Sardina et al. 2018) respectively. 

A second concern is the effects are on two metabolic and pro-apoptotic genes not associated with myeloid malignancies. There should be some discussion and analysis of which RUNX1 and CEBPA mutations affect TET2 binding and recruitment and hypermethylation of BIK or OSBPL5. It would be more relevant to focus on the HOXA9/10 genes. IT would be more novel to identify understudied genes in myeloid neoplasia -- perhaps OSBPL5 -- but there needs to be some mechanistic link between its silencing and leukemogenesis.

Response:

It has been shown that OSBPL5 can be affected in a subtype of AML (Mondet et al. 2021). Significant reduction of expression of the genes involved in mitochondria–ER complexes (mitochondria-associated endoplasmic reticulum membrane, MAM, including OSBPL5) is observed in AML patients with the ASXL1 mutation. 

While BIK appears to be a prominent target for anti-cancer drugs also been used as a therapeutic molecule to treat cancers resistant to standard chemotherapy (reviewed in  (Chinnadurai, Vijayalingam, and Rashmi 2008)

We added the following text to the paper:

Results:

Down-regulation of OSBPL5 has been previously reported in a subtype of AML. While downregulation of BIK was observed in multiple cancers.

Discussion:

Moreover, we demonstrated the reactivation of the expression of pro-apoptotic protein BIK (Bcl-2-interacting killer) by HMA in AML cells. We also showed that hypermethylaiton of BIK might have an effect on the long term patient survival. It is in line with the previously proposed idea that therapeutic approaches to activate the pro-apoptotic BH3-only genes including BIK mingth improve the clinical outcome of chemotherapy treatments in drug-resistant AML.

Round 2

Reviewer 2 Report

This study still lacks critical detail to interpret the results and to support the authors conclusions.

The study defines regions like CpG TL but does not provide information for the interpretation of these sites, and authors do not distinguish nor describe the differences or significance of the results presented in fig 1 a -h. Similarly, RUNX1/CEBPA (TFBS)are derived from HOCOMOCO database and not cell type specific binding sites from ChIP-seq in AML. Authors should provide more information about the number of TFBS analyzed in the dataset, as well as to compare the sites analyzed to RUNX1 ChIP-seq sites from AML cells. Further, what is meant by proximity to TFBS. The methods description is not clear or complete.

In section 2.2, authors attempt to correlate RUNX1 status with chemotherapy responses. Authors describe chemotherapy sensitization yet the 'chemotherapy' treatment description is lacking and Figure . Also, the observed correlation of cytarabine sensitivity and RUNX1 status of cell lines likely due to many factors. More direct support of this could be derived expressing WT RUNX1 and measure shift in sensitivity.

Figure legends still lack sufficient detail. For example, the Figure 2 legend should define the assay (ChIP-qPCR) and provide a more interpretable description of data in the results section (what regions are being targeted and why these were selected). 

Author Response

This study still lacks critical detail to interpret the results and to support the authors conclusions.The study defines regions like CpG TL but does not provide information for the interpretation of these sites, and authors do not distinguish nor describe the differences or significance of the results presented in fig 1 a -h.

Response: First of all, we added the list of the terms and their description. We modified the text and added extra violin plots based on the same data to the supplementary material (Supplementary figure S1). 

The following text in Rusults has been modified to address the reviewer’s comment:

CpGs positions within the close proximity to RUNX1/CEBPA binding sites (TFBS) show an increase in DNA methylation in AML patients with corresponding mutations in comparison to patients without mutations (Fig. 1a,b,d,e,g,h). For the majority of CpGs, only a mild gain in methylation in RUNX1-mutated AML patients has been observed (Fig. 1a, Supplementary Fig. 1a, average ?meth = 0.06), while for the differentially methylated CpG in close proximity to RUNX1 TFBS, the gain in DNA methylation is significantly higher (Fig. 1a, Supplementary Fig. 1a, average ?meth = 0.18). Similar but milder tendency has been observed for the patients with mutations in CEBPA: the gain in DNA methylation genome-wide is significantly lower than that near CEBPA TFBS (Fig. 1g, Supplementary Fig. 1c, average ?meth = 0.09 vs average ?meth = 0.13). The tendency did not stand for CEBPA TFBS near hypermethylated CpG TL but the dififrent is insignificant due to low number of such CpGs (Fig. 1k).

CEBPA mutation leads to a higher level of genome-wide hypermethylation (Fig.1g, grey dots, average ?meth = 0.09) as compared to RUNX1 mutation (Fig. 1a, grey dots, average ?meth = 0.06). In case of the RUNX1 mutation, genome-wide hypermethylation is almost lost for CpG TL (Supplementary Fig. 1b, grey violines) - functional CpG sites associated with changes in gene expression (see Methods), while in the case of CEBPA mutation CpG TL tend to demonstrate hypermethylation independent of the presence of CEBPA TFBS (Supplementary Fig. 1d, grey violines). The mechanism of genome-wide hypermethylation of functional CpGs in patients with CEBPA mutation is unclear, since the majority of genomic CpGs do not have CEBPA TFBS nearby. It suggests that CEBPA regulation of the associated genes might be indirect or specific to a subpopulation of patients as shown in the work of Figueroa (Figueroa et al. 2010).  

Similarly, RUNX1/CEBPA (TFBS) are derived from HOCOMOCO database and not cell type specific binding sites from ChIP-seq in AML. Authors should provide more information about the number of TFBS analyzed in the dataset, as well as to compare the sites analyzed to RUNX1 ChIP-seq sites from AML cells. Further, what is meant by proximity to TFBS. The methods description is not clear or complete. 

Response: 

We consider regions +/- 100bp around CpG as a proximal to TFBS. The corresponding information is available in Methods. Prediction was made using positional weight matrices (PWM) from HOCOMOCO v11 (p-value<0.001) (Kulakovskiy et al. 2018).  We predicted 53443 sites for RUNX1 and 47157 for CEBPA. To reduce the number of false positives we filter out those predictiod TFBS that did not co-locate with the ChIP-seq peak for a corresponding TF from Сistrome (A,B and C categories only)  (Yevshin et al. 2019). After ChIP-seq validation we got 10570 sites for RUNX1 and 5263 for CEBPA respectively.  

We added the following text to the Methods:

TFBS prediction

RUNX1 and CEBPA TFBS were annotated in all CpG centered 200bp regions (+/- 100bp) using positional weight matrices (PWM) from HOCOMOCO v11 (p-value<0.001) (Kulakovskiy et al. 2018). We predicted 53443 sites for RUNX1 and 47157 for CEBPA. To reduce the number of false positives we filter out those predicted TFBS that did not co-locate with the ChIP-seq peak for a corresponding TF from Сistrome (A,B and C categories only) (Yevshin et al. 2019). As a result we obtained 10570 sites for RUNX1 and 5263 for CEBPA respectively.  

In section 2.2, authors attempt to correlate RUNX1 status with chemotherapy responses. Authors describe chemotherapy sensitization yet the 'chemotherapy' treatment description is lacking and Figure . Also, the observed correlation of cytarabine sensitivity and RUNX1 status of cell lines likely due to many factors. More direct support of this could be derived expressing WT RUNX1 and measure shift in sensitivity. 

Response:

We are thankful to the Reviewer for the suggestion. Indeed, the genetic approach of restoring RUNX1 in cells lacking RUNX1 completely may provide a much stronger support for our hypothesis. Unfortunately, we were not able to find such AML cell line. In cells, we used in this study, the RUNX1 gene is mutated and have a premature stop codon in one allele and a substituted amino acid in another. The presence of a mutant form of RUNX1 in cells could counteract the action of WT RUNX1 and will not allow to observe the significant effect on sensitivity to chemotherapy.

Figure legends still lack sufficient detail. For example, the Figure 2 legend should define the assay (ChIP-qPCR) and provide a more interpretable description of data in the results section (what regions are being targeted and why these were selected).

Response: 

We modified figure legends, so now they are more self-descriptive. We also added the assay name to Fig.2. Regions for ChIP-qPCR were selected according to bioinformatics analysis. We focused on genes with the most strikingly changed methylation and expression levels in the case of TF mutation (FC>2, FDR<0.05, absolute expression value > 0.5). We found 12 and 11 genes that meet these criteria for AML patients with RUNX1 and CEBPA mutation respectively, so probes associated with OSBLP5, BIK and HOX9 selected based on this analysis. Probes sequences are provided in Supplemental Table 2.

We modified the following text in the Results:

Next, we focused on genes with the most strikingly changed methylation and expression levels in the case of TF mutation (FC>2, FDR<0.05, absolute expression value > 0.5). BIK/OSBPL5 and HOXA9 are among those genes in the case of RUNX1 and CEBPA mutation respectively. ChIP-qPCR confirms reduced TET2 presense in OSBPL5 and BIK genes in the OCI-AML5 cell line (a line with a reported RUNX1 mutation (Fig. 2a,b)), and in HOXA9 in the Kasumi-6  cell line (a line with a reported CEBPA mutation (Supplementary Fig. S6)) supporting the role of TET2 in TFBS demethylation.

Round 3

Reviewer 2 Report

The revised manuscript is acceptable.